# Preoperative Serum Macrophage Migration Inhibitory Factor Level Correlates with Surgical Difficulty and Outcome in Patients with Autoimmune Thyroiditis

**DOI:** 10.3390/jcm10184034

**Published:** 2021-09-07

**Authors:** Kian-Hwee Chong, Ming-Hsun Wu, Chuang-Wei Chen, Tsung-Han Hsieh, Chieh-Wen Lai

**Affiliations:** 1Division of General Surgery, Department of Surgery, Taipei Tzu Chi Hospital, Buddhist Tzu Chi Medical Foundation, New Taipei City 231, Taiwan; ckianhwee@gmail.com; 2Department of Surgery, National Taiwan University Hospital, Taipei 100, Taiwan; larrywu@hotmail.com; 3Division of Colon and Rectal Surgery, Department of Surgery, Taipei Tzu Chi Hospital, Buddhist Tzu Chi Medical Foundation, New Taipei City 231, Taiwan; Cathy166@hotmail.com; 4School of Medicine, Tzu Chi University, Hualien 970, Taiwan; 5Department of Research, Taipei Tzu Chi Hospital, Buddhist Tzu Chi Medical Foundation, New Taipei City 231, Taiwan; b87404037@gmail.com

**Keywords:** autoimmune thyroiditis, difficulty, outcome, serum biomarker, thyroidectomy

## Abstract

Surgical treatment for autoimmune thyroid disease is theoretically risky due to its chronic inflammatory status. This study aimed to investigate the correlation between preoperative serum migration inhibitory factor (MIF) levels and the difficulty of thyroidectomy in patients with autoimmune thyroiditis. Forty-four patients (average age: 54 years) were prospectively recruited: 30 with autoimmune thyroiditis and 14 with nodular goiter. Preoperative serum samples were collected to measure MIF levels. The difficulty of thyroidectomy was evaluated using a 20-point thyroidectomy difficulty scale (TDS) scoring system. The potential correlations between MIF levels and clinicopathological features as well as postoperative complications were analyzed. Preoperative serum thyroid-stimulating hormone (TSH), TSH receptor antibody, thyroid peroxidase antibodies levels, TDS score, and serum MIF levels were significantly higher in the autoimmune thyroiditis group than those in the goiter group. MIF levels were significantly associated with postoperative transient recurrent laryngeal nerve injury and hypoparathyroidism. MIF levels were positively correlated with TDS score, operation time, and blood loss in the autoimmune thyroiditis group. Increased preoperative serum MIF levels are associated with higher TDS scores, operation time, blood loss, and postoperative complications. Preoperative serum MIF level may be a useful predictor of difficult thyroidectomy and help surgeons provide better preoperative management.

## 1. Introduction

Thyroidectomy has a high mortality rate and was prohibited by the French Medical Society during the late 19th century. Theodor Kocher attained a low mortality rate for thyroid surgery and won a Nobel Prize in 1909 [1]. To date, thyroidectomy is one of the most common surgical procedures performed worldwide.

Autoimmune disease of the thyroid is a frequent cause of thyroiditis and inflammatory thyroid disorders. The most common pathohistological findings in Hashimoto’s thyroiditis are lymphocytic infiltration, follicular destruction, and apoptosis of thyroid epithelial cells. Hashimoto’s thyroiditis often induces a chronic inflammatory status to the thyroid tissue with or without goiter formation, and the thyroid gland may become diffusely enlarged or shrunken. The common pathohistological findings in Graves’ disease include follicular hyperplasia, intracellular colloid droplets, and non-homogenous lymphocytic infiltration, which is predominantly T-lymphocytes. The surgical indications for Graves’ disease and Hashimoto’s thyroiditis are similar, such as suspected malignancy, symptoms of compression, Graves’ ophthalmopathy, and intolerance to long-term medication. Thyroidectomy for Graves’ disease or Hashimoto’s thyroiditis is more difficult than that for nodular goiters and often has a longer operation time and a higher mobility rate [2].

The thyroidectomy difficulty scale (TDS) score was developed by Schneider et al. to measure the degree of surgical difficulty [3,4]. The scale contains four parameters: vascularity, friability, mobility, or fibrosis, and gland size. Using this scale, surgeons can objectively measure the difficulty of thyroidectomy and may predict the risk of postoperative complications. Although TDS has a positive correlation with surgical difficulty and postoperative complications, there are still some limitations with this scoring system, especially the scoring bias between different surgeons. Therefore, we attempted to identify a relatively objective preoperative biomarker and to scrutinize the correlation with TDS in different surgical difficulties of thyroidectomy.

Macrophage migration inhibitory factor (MIF) is a multifunctional proinflammatory cytokine that was initially known as a lymphokine associated with delayed-type hypersensitivity. MIF is thought to induce the inhibition of migration and aggregation of macrophages at an inflammatory lesion [5]. MIF is associated with various autoimmune diseases, such as rheumatoid arthritis [6], and inflammatory bowel disease (IBD) [7]. Moreover, the severity of autoimmune thyroid disease has also demonstrated a relationship with MIF gene polymorphisms (rs755622 single nucleotide polymorphism [SNP]) [8].

Although previous studies have shown a correlation between serum MIF and autoimmune thyroiditis, the clinical application of MIF in thyroidectomy is still lacking. This study aimed to evaluate whether preoperative serum MIF levels correlated with TDS and could be predictive of surgical difficulty in thyroidectomy.

## 2. Patients and Methods

### 2.1. Study Population

The study was approved by the Institutional Review Board of the Buddhist Tzu Chi General Hospital (No. 07-XD-101) and was carried out between January 2019 and May 2020. We prospectively recruited 44 patients (30 patients with autoimmune thyroiditis and 14 patients with multiple nodular goiter) who were scheduled for total/nearly total thyroidectomies at Taipei Tzu Chi Hospital, New Taipei City, Taiwan. The surgery was performed by the same surgeon (Dr. Lai CW). Autoimmune thyroiditis was diagnosed preoperatively by blood tests for abnormal antibodies, including thyroid peroxidase antibodies (TPO antibodies) and thyroid-stimulating hormone (TSH) receptor antibodies (TSH receptor Ab). To avoid other conditions that could alter the serum levels of macrophage MIF, the following patients were excluded: patients with other known autoimmune or inflammatory diseases, pregnant patients, and those with malignant diseases.

Perioperative clinicopathological outcomes, including operation time, blood loss, and postoperative complications, including recurrent laryngeal nerve (RLN) injury, hypoparathyroidism, hematoma, and wound infection were recorded. Hypoparathyroidism was defined as a serum intact PTH level of <10 pg/mL. Postoperative RLN injury or hypoparathyroidism was classified as transient if they recovered within 3 months and permanent if they persisted beyond 3 months.

Immediately after the operation, the 20-point TDS score was completed by a single surgeon (Dr. Lai). The difficulty scale consisted of four variants: vascularity, friability, fibrosis, and gland size. Each item was graded on a scale of 1–5. A maximum score of 20 indicated the most difficult thyroidectomy.

Peripheral blood samples for soluble macrophage MIF levels were obtained from all patients just before their operations. Serum was collected and stored in aliquots at −80 °C for further batch use. No patients received preoperative chemotherapy, radiotherapy, or blood transfusions. The levels of serum MIF were determined quantitatively using human MIF quantikine ELISA kits PDMF00B (R&D Systems, Minneapolis, MN, USA), and the protocols were conducted in accordance with the manufacturer’s instructions.

### 2.2. Statistical Analysis

Data are presented as median ± interquartile range (IQR) and number (%) for continuous and categorical variables, respectively. Continuous variables were assessed using the Mann-Whitney U test. Categorical variables were analyzed using the chi-square test or Fisher’s exact test. The correlation between serum MIF levels and surgical difficulty factors was evaluated using Spearman’s rank correlation analysis. All statistical analyses were performed using the SPSS statistical software (version 18.0, IBM SPSS Inc., Chicago, IL, USA), and statistical significance was set at *p*-values of <0.05.

## 3. Results

### 3.1. Patient Characteristics

A total of 44 patients (33 women and 11 men; 30 with autoimmune thyroiditis and 14 with nodular goiter) were recruited for this study. The baseline patient characteristics are summarized in Table 1.

The median age was 54 years (range 40–64 years) and this was similar in both groups. Preoperative serum TSH [1.564 (0.95–2.90) vs. 1.05 (0.85–1.70)], TSH receptor Ab [10.5 (6.5–16.25) vs. 5 (5–8)] and Anti-TPO Ab [4573 (1563–9405) vs. 28 (17.5–28)] levels are significantly higher in the autoimmune thyroiditis group than the goiter group. The distribution of serum TSH, TSH-receptor Ab, Anti-TPO Ab, and MIF levles in both groups is shown in Figure 1.

Preoperative serum MIF levels and surgical difficulty factors in patients with autoimmune thyroiditis and benign nodular goiters.

The preoperative serum MIF level, TDS score, operation time, and intraoperative blood loss in the two groups are shown in Table 2. Preoperative serum MIF levels [15.91 (4.16–31.64) vs. 3.82 (1.55–6.54), *p* = 0.001] were significantly higher in the autoimmune thyroiditis group than in the goiter group. The TDS score [14 (12–15) vs. 8.5 (6–10), *p* < 0.001] was also significantly higher in the autoimmune thyroiditis group than that in the goiter group. Operation time was longer in the autoimmune thyroiditis group than in the goiter group [75 (61.75–85) min vs. 59 (50–65) min, *p* = 0.006). Intraoperative blood loss [43.50 (28.75–61.75) mL vs. 18.50 (9–46) mL, *p* = 0.03) was also higher in the autoimmune thyroiditis group than in the goiter group.

### 3.2. Correlation from Preoperative Serum MIF Levels to Surgical Complications

The correlation between preoperative serum MIF levels and surgical complications, including RLN injury, hypoparathyroidism, and hematoma, was analyzed using the Mann-Whitney U test. Regarding RLN injury cases, two cases were recorded in the autoimmune thyroiditis group and the patients spontaneously recovered 1 month later. There was a significant correlation between RLN injury and preoperative serum MIF levels [51.37 (47.53–55.21) vs. 6.92 (3.29–18.49), *p* = 0.019] (Table 3). Concerning hypoparathyroidism, there were ten cases in the autoimmune group and three cases in the goiter group. The blood calcium levels returned to normal within 3 months. The preoperative serum MIF levels showed a significant correlation with postoperative hypoparathyroidism [29.24 (10.76–50.82) vs. 4.37 (2.77–47.53), *p* < 0.001]. Additionally, increased TDS score had a positive correlation with postoperative hypoparathyroidism [15 (11–16.5) vs. 12 (9–14), *p* = 0.06] (Table 4). No patient experienced postoperative hematoma in either group.

### 3.3. Correlation between Preoperative Serum MIF Levels and Surgical Difficulty Factors in Patients with Autoimmune Thyroiditis

We further demonstrated the effectiveness of preoperative MIF in determining the difficulty of thyroidectomy in patients with AITD. Using Spearman’s rank correlation analysis, we analyzed the correlation between preoperative serum MIF levels and surgical difficulty factors, including TDS, operation time, and blood loss (Table 5) in patients with thyroiditis. Concerning operation time (*p* = 0.680, *p* < 0.001) and blood loss (*p* = 0.692, *p* < 0.001), there was a significant positive correlation with serum MIF levels. Increased preoperative serum MIF levels are associated with longer operation times and more blood loss. The TDS score also had a significant positive correlation with operation time and blood loss. Nevertheless, the relationship between MIF levels and TDS was positively correlated (*p* = 0.725, *p* < 0.001). Increased MIF levels were associated with higher TDS scores.

## 4. Discussion

The results of the current study demonstrated that preoperative serum MIF levels were significantly elevated in patients with autoimmune thyroiditis. In addition, increased preoperative serum MIF levels were associated with increased TDS score, operation time, blood loss, and postoperative complications, including transient RLN injury and hypoparathyroidism.

MIF is found in epithelial cells and is involved in several innate and adaptive immune responses, including cell-mediated immune responses, immune regulation, and inflammation. Overexpression of MIF induces an increase in macrophage cytokine levels, whereas MIF deficiency decreases cytokine secretion and increases fibroblast adipogenesis [5,9,10,11,12,13]. Some studies have shown that MIF is thought to be involved in thyroid diseases [14]. The location of the MIF gene is found at 22q11.2, and the two polymorphisms rs5844572 [15] and rs755622 [16] are thought to correlate with the severity of inflammatory disorders [17,18,19,20,21,22,23,24]. Graves’ disease and Hashimoto’s disease are the most common types of autoimmune thyroiditis and they may cause hyperthyroidism and hypothyroidism, respectively. Previous studies have also shown a significant relationship between MIF in Graves’ disease and Hashimoto’s disease [8,25,26,27]. Liu et al. demonstrated that MIF and CD74 are risk factors for Graves’ disease and Graves ophthalmopathy [25]. Liu et al. also showed that the distribution of the C allele, especially the C/C genotype, of the rs755622 SNP in MIF might be a risk factor for developing goiter [8]. The other two studies showed that MIF levels are higher in Hashimoto’s disease, and MIF is also involved in the development of autoimmune thyroid disease [26,27]. In the current study, our data suggested that preoperative serum MIF levels were significantly higher in these two autoimmune thyroiditis groups than those in the control group [15.91 (4.16–31.64) vs. 3.82 (1.55–6.54); *p* = 0.001]. We speculate that the serum level of MIF may be a predictor of the inflammatory status of thyroid diseases. However, the relationship between the severity of thyroiditis and serum level of MIF requires further investigation.

Hashimoto’s thyroiditis is the most common cause of hypothyroidism in non-iodine-deficient countries. The most common pathohistological findings in Hashimoto’s thyroiditis include lymphocytic infiltration, follicular destruction, and apoptosis of thyroid epithelial cells. It often induces a chronic inflammatory status to the thyroid tissue with or without goiter formation, and the thyroid gland may become diffusely enlarged or shrunken. Hashimoto’s disease is often controlled by medical treatment; however, surgical treatment is indicated when malignant tumors are proven or highly suspected, and compressive symptoms develop [28]. Graves’ disease is an autoimmune disease and the most common cause of hyperthyroidism. The common pathohistological findings in Graves’ disease include follicular hyperplasia, intracellular colloid droplets, and non-homogenous lymphocytic infiltration, which is predominantly T-lymphocytes. The indications for surgery in Graves’ disease include medical treatment failure or relapse, pregnancy or planning for pregnancy, large goiter with compressive syndrome, suspected or proven for malignancy, patients under the age of 18 years, Graves’ ophthalmopathy, and patient’s preference [28]. In our study, the indication for surgery in these two autoimmune thyroid diseases was the presence of a compressive tumor or suspected malignancy. All surgical procedures were total or near-total thyroidectomy. To eliminate the interference in serum MIF levels, the final pathological diagnosis of thyroid malignancy was excluded from this study.

The surgical management of autoimmune thyroid disease is challenging, even for experienced surgeons. Chronic inflammation may lead to fibrosis of the thyroid gland and surrounding structures, thus increasing the difficulty of thyroidectomy and resulting in more postoperative complications. McManus et al. demonstrated that the overall surgical complications were significantly higher in the Hashimoto’s thyroiditis group (15.1%) than those in the non-Hashimoto’s thyroiditis group (8.8%) [29]. In detail, the rate of transient complications in the Hashimoto’s thyroiditis group was 11.9%, compared with 6.8 % in the non-Hashimoto’s thyroiditis group, and the rate of permanent complications in the Hashimoto’s thyroiditis group was 2.6% and 0.3% in the non-Hashimoto’s thyroiditis group. All results were statistically significant. In the current study, our data demonstrated that thyroidectomy for patients with autoimmune thyroiditis had a longer operation time [75 (61.75–85) min vs. 59 (50–65) min] than that for goiter (*p* = 0.006). Two of the 30 patients with autoimmune thyroiditis (6.6%) experienced transient RLN injury, compared with no (0.0%) transient RLN injury in the 14 patients without autoimmune thyroiditis (0.0%). In addition, 10 of the 30 patients with autoimmune thyroiditis (33.3%) had transient hypoparathyroidism, compared with 3 of the 14 patients without autoimmune thyroiditis (21.4%). No postoperative hematoma or permanent complications were observed. In the current study, we demonstrated that increased preoperative serum MIF levels were significantly associated with surgical complications, including transient RLN injury and hypoparathyroidism. Furthermore, increased TDS score had a positive correlation with postoperative hypoparathyroidism.

Since the surgical treatment for autoimmune thyroid disease is theoretically riskier, adequate preoperative decision making for the surgery, including the operation method, the need for intraoperative nerve monitoring or hemostatic agents, and informed consent from the patient are more important. In addition, perioperative management, such as preventive parathyroid gland auto-transplantation should be considered. To identify the risk of thyroidectomy, Schneider et al. developed a novel TDS and it has been adopted worldwide [4]. Their study showed that the scale is correlated with longer operation times and higher complication rates. Another study by Saadi et al. also demonstrated that the TDS score of the autoimmune thyroiditis group was higher than that of the control group and was correlated with longer operation times and longer hospital stay [30]. In our study, we found that the TDS score was significantly higher in the thyroiditis group than in the goiter group. Moreover, the TDS score was positively associated with longer operation time, more blood loss, and postoperative transient hypothyroidism. Our results are in accordance with those of previous studies [4,30].

Although TDS has a positive correlation with surgical difficulty and postoperative complications, there are still some limitations in this scoring system, especially the scoring bias between different surgeons. Therefore, we attempted to identify a relatively objective preoperative biomarker and to scrutinize the correlation with TDS in different surgical difficulties of thyroidectomy. To the best of our knowledge and according to literature review, few studies have focused on the relationship and application of preoperative predictors in difficult thyroidectomy. Schneider et al. demonstrated that preoperative serum thyroglobulin and antithyroglobulin antibodies may be predictors of difficult thyroidectomy and are associated with longer operation time and more complications [3]. Our study suggests that preoperative serum MIF level is positively associated with longer operation time, more blood loss and higher TDS score, and postoperative complications, including transient RLN injury and hypoparathyroidism in patients. Although TDS can also help surgeons to determine the difficulty of thyroidectomy, it remains controversial in this scoring system. In the current study, the operation was performed by a single surgeon, and our results demonstrated that preoperative MIF levels have a positive correlation with the TDS. Therefore, we speculate that preoperative serum MIF level may be a useful tool in predicting and determining the difficulty of thyroidectomy preoperatively. It also provides more information to surgeons for better preoperative preparation and arrangement, especially in patients with autoimmune thyroiditis.

There are some limitations in the current study. First, the sample size was relatively small. A larger study size is needed to further investigate the postoperative complications. Second, we did not study postoperative serum MIF levels and observe whether the level declined or increased after surgery. Third, the effect of the combination of MIF and TDS to evaluate the difficulty of thyroidectomy was not evaluated in this study. Therefore, a larger sample study with a combination of serum MIF levels and TDS is needed in the future to better define the clinical use of serum MIF in difficult thyroidectomy.

In conclusion, preoperative serum MIF level is significantly elevated in patients with autoimmune thyroiditis. In addition, increased preoperative serum MIF levels were associated with higher TDS scores, operation time, blood loss, and postoperative complications, including transient RLN injury and hypoparathyroidism. Preoperative serum MIF levels may be a useful predictor of difficult thyroidectomy. This information can help surgeons to provide better preoperative management, including the need for intraoperative nerve monitoring, hemostatic agents, risk counseling, and efficient operation schedules. In addition, prophylactic parathyroid gland auto-transplantation may be considered in risky thyroidectomy.

## Figures and Tables

**Figure 1 jcm-10-04034-f001:**
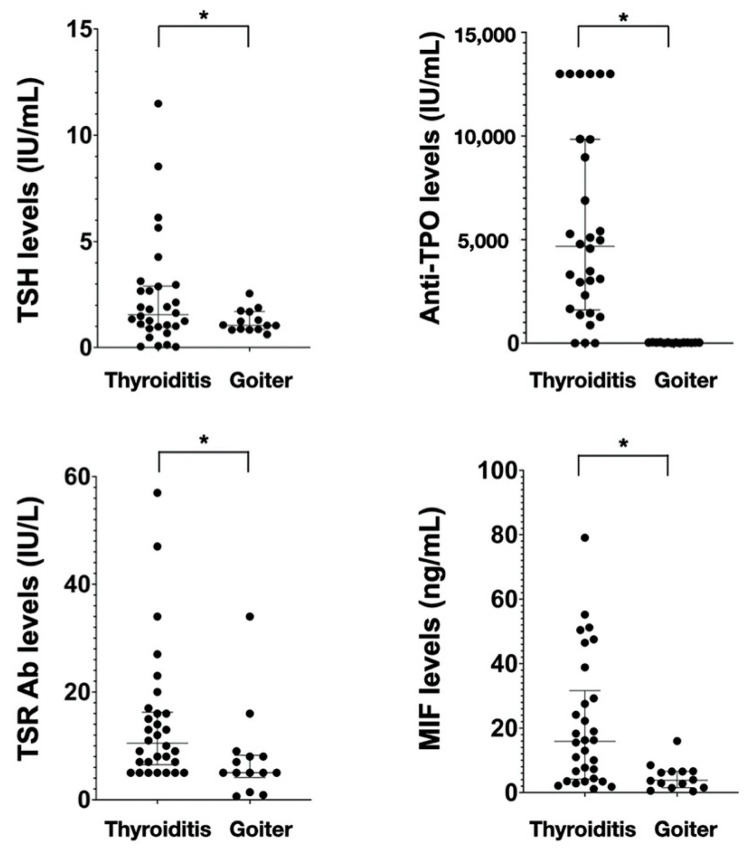
Serum levels of TSH, TSH-receptor Ab, anti-TPO Ab, and MIF in different groups. Values are presented as median ± IQR. TSR Ab, TSH-receptor antibody; * Statistically significant difference.

**Table 1 jcm-10-04034-t001:** Clinicopathological characteristics of patients.

	Group		*p*
	Thyroiditis (*n* = 30)	Goiter (*n* = 14)	
Gender			0.312
Female	23 (76.7%)	10 (71.4%)	
Male	7 (23.3%)	4 (28.6%)	
Age (years)	51.50 (38–61.50)	51 (40–57)	0.633
Anti-TPO Ab (IU/mL) ^a^	4573 (1563–9405)	28 (17.5–28)	<0.001 *
TSH-receptor Ab (IU/L)	10.5 (6.5–16.25)	5 (5–8)	0.0017 *
TSH (IU/mL) ^b^	1.564 (0.95–2.90)	1.05 (0.85–1.70)	0.002 *

Values are presented as median ± IQR. ^a^ Anti-TPO Ab, antithyroid peroxidase antibody; ^b^ TSH, thyroid-stimulating hormone; * Statistically significant difference.

**Table 2 jcm-10-04034-t002:** Correlation of MIF, TDS, OP time, and blood loss in different groups.

	Group		*p*
	Thyroiditis (*n* = 30)	Goiter (*n* = 14)	
MIF (ng/mL)	15.91 (4.16–31.64)	3.82 (1.55–6.54)	0.001 *
TDS	14 (12–15)	8.5 (6–10)	<0.001 *
OP time (min)	75 (61.75–85)	59 (50–65)	0.006 *
Blood loss (mL)	43.50 (28.75–61.75)	18.50 (9–46)	0.030 *

Values are presented as median ± IQR. MIF, macrophage migration inhibitory factor; TDS, thyroidectomy difficulty scale; OP, operation; * Statistically significant difference.

**Table 3 jcm-10-04034-t003:** Correlation between RLN injury and MIF or TDS.

	RLN Injury		*p*
	No (*n* = 42)	Yes (*n* = 2)	
MIF (ng/mL)	6.92 (3.29–18.49)	51.37 (47.53–55.21)	0.019 *
TDS	12 (9.75–14)	15 (14–16)	0.152

Values are presented as median ± IQR. MIF, macrophage migration inhibitory factor; TDS, thyroidectomy difficulty scale; RLN, recurrent laryngeal nerve; * Statistically significant difference.

**Table 4 jcm-10-04034-t004:** Correlation between postoperative hypoparathyroidism and MIF or TDS.

	Hypoparathyroidism		*p*
	No (*n* = 31)	Yes (*n* = 13)	
MIF (ng/mL)	4.37 (2.77–47.53)	29.24 (10.76–50.82)	<0.001 *
TDS	12 (9–14)	15 (11–16.5)	0.006 *

Values are presented as median ± IQR. MIF, macrophage migration inhibitory factor; TDS, thyroidectomy difficulty scale; * Statistically significant difference.

**Table 5 jcm-10-04034-t005:** Correlation between MIF and TDS/OP time/Blood loss.

	MIF	TDS	OP Time	Blood Loss
MIF				
TDS	0.725 ***			
OP time	0.680 ***	0.628 ***		
Blood loss	0.692 ***	0.687 ***	0.520 **	

Statistical analysis was performed using Spearman’s rank correlation analysis. MIF, macrophage migration inhibitory factor; TDS, thyroidectomy difficulty scale; OP, operation. ** *p* < 0.01, *** *p* < 0.001.

## Data Availability

The datasets used and/or analyzed during the current study are available from the corresponding author on reasonable request.

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
