# Peer review of "Preoperative Serum Macrophage Migration Inhibitory Factor Level Correlates with Surgical Difficulty and Outcome in Patients with Autoimmune Thyroiditis"

_jcm, 2021, doi:10.3390/jcm10184034_

Round 1

Reviewer 1 Report

Autoimmune thyroid disease (AITD) remains very common. Some literature data are showing that it affects 2 – 5 % of the population, with women being affect more frequently due to a different hormonal background. Thyroidectomy still remains as one the most effective tool in AITD treatment. In some countries, thyroidectomy is used majorly only on the late stages of AITD.

The manuscript brings very important topic which is increasing quality of the outcome for AITD patient after thyroidectomy by prognosing level of surgery difficulty implementing a new marker - macrophage migration inhibitory factor (MIF). The big drawback of this article – very small number of patients enrolled in this research (14 patients with ongoing AITD and 14 patients with nodular goitre), which does not allow to do sufficient statistical analysis and to make some significant conclusions. Conclusion on that the MIF level is higher in patients with ongoing AITD than in patients with goitre is obvious and logical, as in the second variant there is no left at all from thyroid gland’s normal tissue. With this conclusion is only possible to indicate that every AITD patient has high MIF level and has high thyroidectomy difficulty scale (TDS) score and does not help to distinguish surgery difficulty of AITD patients – all cases would be difficult, and it is not necessary to use additional marker. This article does not provides sufficient information about evaluation of MIF level within the AITD patients’ group, which is related with very small study group. In case of small patients’ group should change the study design, so it would be possible to present results in other way to be more helpful in evaluation of situation. For example, evaluation of MIF levels before and after surgery and patient’s recovery quality would reveal role of MIF level – as marker better, and give some sufficient information how to follow up the patient after thyroidectomy.

Another weakness of the manuscript is statistical analysis.  All data were performed as mean ± standard deviation (SD), but big SD values in TSH, MIF and auto-antibodies levels indicates that data are not normally distributed. Therefore, all data, which are not normally distributed, should be presented as median ± IQR. Also, it is not clear why authors used parametric values (mean ± SD) for the Mann-Whitney U test, which is used for mainly non-parametric analysis. Use of figures for the TSH, MIF and auto-antibodies level results’ presentation (with dot blot) would improve ‘Results’ section, by showing disperse of the data better.

 Recommendations for manuscript improvement:

  • to enlarge the patients’ group minimum till 30 AITD patients which will improve statistics quality and would allow better to show results of different thyroidectomy difficulties and MIF levels;
  • to add results’ analysis including MIF measurement after thyroidectomy.

Author Response

Reviewer 1:

Comments and Suggestions for Authors

Recommendations for manuscript improvement:

  1. All data were performed as mean ± standard deviation (SD), but big SD values in TSH, MIF and auto-antibodies levels indicates that data are not normally distributed. Therefore, all data, which are not normally distributed, should be presented as median ± IQR.

Response: Thank you for the comment. We have revised all our data to present as median ± IQR.

  1. Use of figures for the TSH, MIF and auto-antibodies level results’ presentation (with dot blot) would improve ‘Results’ section, by showing disperse of the data better.

Response: Thank you for the comment. We have revised in the result part and used figure (Figure 1) for the TSH, MIF and auto-antibodies level results’ presentation (with dot blot)

  1. To enlarge the patients’ group minimum till 30 AITD patients which will improve statistics quality and would allow better to show results of different thyroidectomy difficulties and MIF levels; or to add results’ analysis including MIF measurement after thyroidectomy.

Response: Thank you for the comment. As we listed the study’s limitation, the sample size was relatively small, and we did not study postoperative serum MIF levels and observe whether the concentration declined or increased after surgery. For the concerns about the sample size of thyroiditis patents, we have followed your suggestion and total 30 AITD and 14 goiter patients those who were already collected for measuring MIF levels in our database were analyzed. We have added them in the result section “Correlation between preoperative serum MIF levels and operative difficulty factors in patients with autoimmune thyroiditis

We want to further demonstrate if preoperative MIF help to distinguish surgery difficulty of AITD patients. Using Spearman rank correlation analysis, we analyzed the correlation between preoperative serum MIF levels and operative difficulty factors, including TDS, operation time, and blood loss (Table 5) in thyroiditis patients. Whether operation time (p =.680, p < 0.001), or blood loss (p =.692, p < 0.001), there was a significant positive correlation with serum MIF levels. Increased preoperative serum MIF levels are associated with longer operation times and more blood loss. The TDS score also had a significant positive correlation with operation time and blood loss. Nevertheless, the relationship between MIF levels and TDS was positively correlated (p = .725, p < 0.001). Increased MIF levels were associated with higher TDS scores.”  This may better provide evidence that preoperative serum MIF level is also a useful predictor for difficult thyroidectomy in thyroiditis patients. We appreciate your great suggestion.

Table 5 Correlation between MIF and TDS/OP time/Blood loss in patients with autoimmune thyroiditis

MIF

TDS

OP time

Blood loss

MIF

TDS

.725***

OP time

.680***

.628***

Blood loss

.692***

.687***

.520**

Statistical analysis was done by Spearman rank correlation analysis. MIF, macrophage migration inhibitory factor; TDS, thyroidectomy difficulty scale; OP, operation. **p < .01, ***p < .001.

Reviewer 2 Report

The manuscript is nicely written. The topic as such is important and relevant to the scientific community as well as to corresponding surgeons.

I would recommend to refine the introduction and discussion.

For instance: Lines 332-337 define the aim of the study much better and should therefore go to the introduction part.

The description of the diseases as such (lines 267-280) may also be better in the introduction.

Overall, the discussion is partly redundant - i. e. lines 320-322.  

Typos:

Table 1: the values for Anti-TPO Ab are mixed up.

Line 168:  ± is shown twice

Line 263: the second bracket is missing. Values are messed up.

Author Response

Reviewer 2:

Comments and Suggestions for Authors

The manuscript is nicely written. The topic as such is important and relevant to the scientific community as well as to corresponding surgeons.

  1. For instance: Lines 332-337 define the aim of the study much better and should therefore go to the introduction part.

Response: Thank you for the comment. For the concerns, we have modified the content in the introduction part to make it clearer “Although TDS has a positive correlation with surgical difficulty and postoperative complications, there are still some limitations in this scoring system, including the bias between different surgeons, can only be performed during the operation period, and lack of preoperative biomarkers. Therefore, we attempted to identify a relatively objective preoperative biomarker and scrutinize the correlation with TDS in different surgical difficulties of thyroidectomy.”

  1. The description of the diseases as such (lines 267-280) may also be better in the introduction.

Response: Thank you for the comment. For the concerns, we have modified the content in the introduction part to make it clearer “The most common pathohistological appearance of Hashimoto’s thyroiditis is lymphocytic infiltration, follicular destruction, and apoptosis of thyroid epithelial cells. It often induces a chronic inflammatory status to the thyroid tissue with or without goiter formation, and the thyroid gland may become diffusely enlarged or shrunken [29]. The common pathohistology of Graves’ disease is follicular hyperplasia, intracellular colloid droplet, and non-homogenous lymphocytic infiltration, which is predominantly T-lymphocytes”

  1. Overall, the discussion is partly redundant - i. e. lines 320-322.  

Response: Thank you for the comment. We have deleted the redundant part as your suggestion.

  1. Typos: Table 1: the values for Anti-TPO Ab are mixed up. Line 168:  ± is shown twice. Line 263: the second bracket is missing. Values are messed up.

Response: Thank you for the comment. We have rechecked the whole manuscript and corrected the errors.

Round 2

Reviewer 1 Report

The manuscript sufficiently has improved. With addition of patients' number and figure, presentation of results has became more clearer, and more significant.  

Author Response

Your valuable comments significantly help us to improve our work. We appreciate your great suggestion.

Reviewer 2 Report

It is nice to see, that the number of patients has increased due to the extension of the time range. Therefore, the value of the manuscript has increased significantly. 

Earlier remarks have been taken care of and changes can be accepted.

Please, check the following sentence in line 111-112: ",can only be performed during the operation period, and lack of preoperative biomarkers."

It seems to be misplaced or or not connected to the prior section.

Figure 1: readability of names on the abscissa and ordinate needs to be improved. The resolution should be increased.

Line 254: replace control with goiter.

Author Response

  1. It is nice to see, that the number of patients has increased due to the extension of the time range. Therefore, the value of the manuscript has increased significantly. Earlier remarks have been taken care of and changes can be accepted.

Response: Your valuable comments significantly help us to improve our work. Thank you for the suggestion.

  1. Please, check the following sentence in line 111-112: ",can only be performed during the operation period, and lack of preoperative biomarkers." It seems to be misplaced or not connected to the prior section.

Response: Thank you for the comment. We have modified this sentence to “Although TDS has a positive correlation with surgical difficulty and postoperative complications, there are still some limitations in this scoring system, especially the scoring bias between different surgeons. Therefore, we attempted to identify a relatively objective preoperative biomarker and to scrutinize the correlation with TDS in different surgical difficulties of thyroidectomy.” in the introduction section.

And “Although TDS has a positive correlation with surgical difficulty and postoperative complications, there are still some limitations in this scoring system, especially the scoring bias between different surgeons. Therefore, we attempted to identify a relatively objective preoperative biomarker and to scrutinize the correlation with TDS in different surgical difficulties of thyroidectomy.” in the discussion section.

  1. Figure 1: readability of names on the abscissa and ordinate needs to be improved. The resolution should be increased.

Response: Thank you for the comment. We have revised the Figure 1. We appreciate your great suggestion.

  1. Line 254: replace control with goiter.

Response: Thank you for the comment. We have replaced control with goiter “Concerning hypoparathyroidism, there were ten cases in the autoimmune group and three cases in the goiter group.”